# Enhanced N$_2$O Emissions from Winter Wheat Field Induced by Winter Irrigation in the North China Plain

Yunhao An [1,2,†], Zhe Gu [1,2,*,†], Xiyun Jiao [1,2,3], Qi Wei [1,2], Junzeng Xu [1,2,3] and Kaihua Liu [1,2]

1 College of Agricultural Science and Engineering, Hohai University (Jiangning Campus), No. 8 Focheng West Road, Nanjing 211100, China; 170402060004@hhu.edu.cn (Y.A.); xyjiao@hhu.edu.cn (X.J.); weiqi@hhu.edu.cn (Q.W.); xjz481@hhu.edu.cn (J.X.); 20200939@hhu.edu.cn (K.L.)
2 State Key Laboratory of Hydrology-Water Resources and Hydraulic Engineering, Hohai University, No. 1 Xikang Road, Nanjing 210098, China
3 Cooperative Innovation Center for Water Safety & Hydro Science, Hohai University, No. 1 Xikang Road, Nanjing 210098, China
* Correspondence: zhegu2017@163.com
† These authors contributed equally to this work.

**Abstract:** Winter irrigation is important for wheat in meeting crop water requirements in spring, but it alters soil moisture dynamics and affects soil N$_2$O production and emission. To assess the effects of winter irrigation on soil N$_2$O emissions in a winter wheat field, an in situ experiment was conducted from 1 October 2019 to 1 March 2020 with one control treatment (CK) and five levels of winter irrigation quantities (irrigated to 60%, 70%, 80%, 90%, 100% of the soil water holding capacity, namely WHC60–WHC100, respectively). The results showed that winter irrigation had an impact on soil N$_2$O emission. The emission peaks were not investigated immediately after winter irrigation, but at two days after, which were increased by 4.3–17.0 µg·m$^{-2}$·h$^{-1}$ under WHC60–100 treatments compared to the CK. The cumulative N$_2$O emissions after winter irrigation from WHC60–100 were 1.1–3.9 times higher than that of CK, indicating that the cumulative N$_2$O emission has an increase trend with the increase of soil water content regulated by irrigations. Pearson correlation analysis showed that the correlation between soil N$_2$O flux and soil temperature were moderate with correlation coefficients of about 0.65. While the correlation between soil N$_2$O flux and soil water content was poor during the investigate winter season with correlation coefficients ranging between 0.08 and 0.25. Future studies should focus on the general N$_2$O emission responses to winter irrigation and environmental factors with the support of experiment data from several winter seasons.

**Keywords:** nitrous oxide (N$_2$O) emission; freezing–thawing; winter irrigation; soil moisture content; soil temperature

## 1. Introduction

The trace gas nitrous oxide (N$_2$O) is one of the most important greenhouse gases, which substantially contributes to global warming and ozone depletion [1]. According to statistics, anthropogenic N$_2$O emissions have been rising, and were 10.6 ± 2.7 Mt N$_2$O yr$^{-1}$ during the period 2007–2016 [2]. Agriculture activities are a major source of N$_2$O emission, accounting for around 78% of anthropogenic N$_2$O emissions globally, and cropland soils emitted around 3 Mt N$_2$O yr$^{-1}$ during the period of 2007–2016 [2]. In addition, previous studies have shown that winter and spring are important periods for soil N$_2$O emission, especially during soil freezing–thawing period [3,4].

Soil freezing–thawing cycle is a phase change phenomenon of soil water, which often occurs in the mid-high latitudes and high elevations. Accompanied with soil water transfer and heat exchange, the freezing–thawing process has a great influence on soil structure [5], nitrogen (N) dynamics and migration [6] and microbial community [7,8], as well as soil N$_2$O emission [9,10]. Many studies show that freezing–thawing cycles stimulated soil

$N_2O$ emissions in different ecosystems [11], including boreal peatlands [12] and alpine meadow [13]. However, some studies reported no apparent $N_2O$ emissions during soil freezing–thawing cycles from an alpine grass soil [14] or from Norway spruce forest soil [15]. These inconsistent results between studies could be attributed to differences in soil water and heat status, soil inorganic nitrogen content and intensity and frequency of freezing-thawing action [16].

In China, the total area of seasonally frozen ground (excluding instantaneous frozen ground) is estimated at $5.36 \times 10^6$ km$^2$, mainly distributing in areas with latitudes higher than $24°$ [17]. The North China Plain, located between $32°$ N and $40°$ N, is a typical seasonal freezing–thawing area as well as an important grain production base with a wheat-maize double cropping system. In the area, the shortage of water resources is very serious and in spring the soil water content could hardly satisfy the requirements of winter wheat at jointing stage. Winter irrigation applied before soil freezing could preserve soil water for spring crop growing, reducing the total quantity of irrigation in the spring peak. Besides, winter irrigation has the advantages of alleviating soil compaction and preventing harmful insect [18].

However, winter irrigation alters soil moisture dynamics and has significant influence on soil $N_2O$ production and emissions. Generally, soil $N_2O$ emission reaches a peak at a soil moisture of 60–70% water filled pore space (WFPS), and saturation condition may suppress $N_2O$ emissions [19,20]. Teepe et al. [21] drew a conclusion through a laboratory experiment that $N_2O$ emissions during soil thawing increased with an increase of WFPS from 42 to 64%, but it decreased between 64% and 76% WFPS in agricultural soils. However, little information so far has been provided on the effect of winter irrigation on the dynamics of soil $N_2O$ emission, and in situ research is lacking. As recorded, the winter of 2019–2020 is the second warmest winter since 1960 in the study area. The warm temperature altered the intensity and frequency of freezing–thawing cycles, affecting $N_2O$ emissions. Therefore, it is necessary to study the dynamics of soil $N_2O$ emissions under winter irrigation treatments during winter with the background of global warming. The objectives of this study were (1) to investigate the dynamics of soil $N_2O$ emission in a warm winter, (2) to investigate the responses of soil $N_2O$ emission to different amounts of winter irrigation, and (3) to analyze the relationship between soil $N_2O$ emission and environmental factors.

## 2. Materials and Methods

### 2.1. Study Sites

The experiment was conducted from 1 October 2019 to 1 March 2020 at Nanpi Eco-Agricultural Experimental Station of Chinese Academy of Sciences, Hebei province, China ($116°40'$ E, $38°00'$ N, 11 m above sea level). The study area has a warm temperate semi-humid continental monsoon climate. The long-term (1953–2018) mean annual air temperature is 12.3 °C, with the lowest monthly temperature of −7.9 °C in January and the highest equals to 22.4 °C in July. Mean annual precipitation is about 480 mm, most of which falls between July and September and only 2% of which falls in winter. The soil in this area generally starts to freeze in November and thaw in March, and the maximum freezing depth is about 30–40 cm (data sourced from the National Meteorological Information Center, http://data.cma.cn (accessed on 1 September 2019)). The soil is classified as chestnut soil and basic physicochemical characteristics are shown in Table 1. For the general soil analysis, soil samples (0–60 cm) were taken from the experiment field with three replications before the experiment began.

**Table 1.** Basic physicochemical characteristics of each soil layer at the study site.

| Soil Layer (cm) | Bulk Density (g/cm³) | Soil Particle Composition | | | Organic C (g/kg) | $NH_4^+$ (mg/kg) | $NO_3^-$ (mg/kg) | Available P (mg/kg) | Exchangeable K (mg/kg) |
|---|---|---|---|---|---|---|---|---|---|
| | | Fraction Clay (<0.002 mm) | Fraction Silt (0.002–0.05 mm) | Fraction Sand (>0.05 mm) | | | | | |
| 0–5 | 1.20 | 5.31 | 74.59 | 20.10 | 15.92 | 1.97 | 3.64 | 8.87 | 53.94 |
| 5–10 | 1.29 | 11.89 | 73.96 | 14.15 | 16.19 | 1.33 | 3.68 | 6.34 | 32.21 |
| 10–20 | 1.40 | 5.56 | 73.98 | 20.46 | 12.56 | 1.29 | 1.35 | 2.68 | 30.52 |
| 20–40 | 1.54 | 4.09 | 70.57 | 25.34 | 10.70 | 1.15 | 0.68 | 2.23 | 29.37 |
| 40–60 | 1.52 | 5.95 | 76.89 | 17.16 | 10.34 | 1.53 | 0.56 | 2.41 | 34.05 |

Note: C, P and K represent carbon, phosphorus and potassium, respectively.

## 2.2. Experimental Design

The experiment included one control treatment (no winter irrigation, CK) and five winter irrigation treatments (WHC60, WHC70, WHC80, WHC90 and WHC100). For the winter irrigation treatments, the root zone soils in 30 cm were irrigated, respectively, to 60%, 70%, 80%, 90%, and 100% of the soil water holding capacity on 15 December 2019. Before winter irrigation, the soil water content was measured, with an average of about 20.0 cm³·cm⁻³. The treatments were replicated three times for a total amount of 18 plots, which were placed in a completely randomized block design (Figure 1). Each plot was 6.0 m × 10.0 m and the plots were separated by ridges (30 cm wide and 25 cm height) and 1 m buffers to minimize the effect of horizontal water movement.

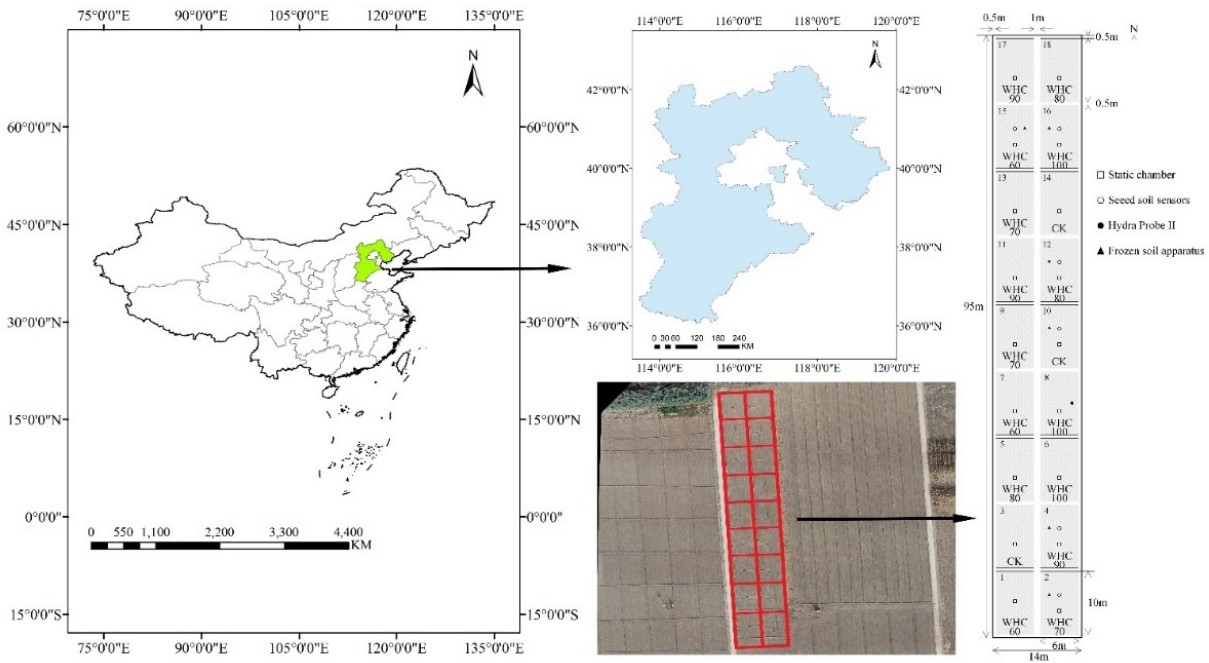

**Figure 1.** Geographic location of the study area and plots arrangements.

The previous maize was harvested on 15 October 2019 and the maize straw was chopped and mixed in the field. The pre-seeding irrigation was applied on 17 October 2019 in order to improve germinating ratio of winter wheat and the irrigation amount was about 60 mm. Inorganic compound fertilizer (N-$P_2O_5$-$K_2O$,18-23-5) was applied at 300kg/ha before pre-seeding irrigation. The winter wheat cultivar "xiaoyan81" was sown on 24 October 2019. Cultural practices, such as pest control, and fertilization were executed following local experience when they were necessary for wheat growth.

## 2.3. Measurements

A meteorological station (Beijing Intell Sun Technology Limited Company, Beijing, China) was set up in an open area near the experiment field to automatically monitor climatic parameters every hour, included air temperature, precipitation, wind speed, etc.

A frozen soil depth measuring instrument (ZY 4000, Henan Zhongyuan Photoelectricity Measurement and Control Technology Co., Ltd., Zhengzhou, China) was installed in one plot with control treatment to measure the changes of soil freezing depth automatically. The soil temperature and moisture sensors (5TE, HSTL-102STRWS, Beijing Huakong Xingye Technology Development Co., Ltd., Beijing, China) were used to monitor the continuous changes of soil temperature and moisture at 5, 10, 20, 40 and 60 cm soil depth in one plot of each treatment, and the data was collected and transmitted automatically per hour. The number of freezing–thawing cycles was identified based on the sensors.

Field water holding capacity was measured using cutting ring method. The undisturbed soil samples from different soil layers were firstly collected using cutting rings. Soak the undisturbed soil samples in water for 24 h, and then place the saturated soil sample on filter paper for 8 h to eliminate surplus gravity water before the soil water content (water holding capacity) was measured by the oven drying method.

### 2.4. Gases Sampling and N<sub>2</sub>O Emissions Calculation

*2.4. Gases Sampling and $N_2O$ Emissions Calculation*

The static chamber–gas chromatography method was used to measure soil $N_2O$ in situ. The sampling interval was usually once every two weeks over winter and additional sampling was conducted 2–3 days after rainfall and irrigation. In each plot, a rectangular stainless-steel base frame (50 cm length, 50 cm width, 20 cm height) was permanently inserted into soil to a depth of 10 cm. For gas sample collection, opaque chambers (50 cm length, 50 cm width, 50 cm height) made of stainless steel were fitted to the base frame. The chambers were wrapped with reflecting tin foil to avoid over-fast temperature increases within the chamber. In order to increase air circulation within the chamber, a fan was mounted on the top of the chamber. A septum was equipped on one side of the chamber for gas sampling and a thermometer was mounted inside the chamber.

Before gas sampling, water was added to the recesses of the base frame to seal the connection between the chamber and base frame to avoid gas exchange. Gas sampling was performed between 9:00 and 12:00 AM on sunny days, during which $N_2O$ emission rates were close to the diurnal averages. At each sampling, four gas samples were collected using a gastight syringe (100 mL) with a three-way valve from inside the chamber at 0, 10, 20, and 30 min after the chamber was put on the base frame. Meanwhile, the temperature in the chamber was recorded. Gas samples were immediately injected into a pre-evacuated 50-mL gas-tight aluminum foil air bag (E-swich, Shanghai Shenyuan Scientific Instrument Co., Ltd., Shanghai, China) and transferred to the lab for further analysis. $N_2O$ concentrations in gas samples were analyzed by gas chromatography (Agilent 7890A, Agilent Technologies, Santa Clara, CA, USA) equipped with an electron capture detector. A standard gas was used to calibrate the measurements from the gas chromatography.

$N_2O$ fluxes were calculated based on the concentration gradient in each chamber and adjusted for air temperature and atmospheric pressure measured at the time as the gas sampling:

$$F = \frac{dc}{dt} \times \frac{M}{V_0} \times \frac{P}{P_0} \times \frac{T_0}{T} \times H \tag{1}$$

where, $F$ represents the $N_2O$ flux ($\mu g \cdot m^{-2} \cdot h^{-1}$), $\frac{dc}{dt}$ represents the slop of linear regression for the $N_2O$ concentration gradient as a function of time, $M$ represents the molecular mass of $N_2O$ ($g \cdot mol^{-1}$), $P$ represents the atmospheric pressure (Pa), $T$ is the absolute temperature (K) during sampling, $V_0$, $T_0$, and $P_0$ are the gas mole volume ($L \cdot mol^{-1}$), absolute air temperature (K) and atmospheric pressure (Pa) under standard condition, respectively, and $H$ is the height of the chamber (cm). A positive $F$ value means that there is a net emission of $N_2O$, and a negative value is opposite [22].

Cumulative $N_2O$ emission ($E$, g ha$^{-1}$) in the field was calculated by the following equation [23]:

$$E = \sum \left\{ \frac{(F_i + F_{i+1})}{2} \times (t_{i+1} - t_i) \times 24 \times 10^{-2} \right\} \tag{2}$$

where $F_i$ represents the $i$th measurement of $N_2O$ flux ($\mu g \cdot m^{-2} \cdot h^{-1}$), $(t_{i+1} - t_i)$ is the days between two adjacent measurements, and $24 \times 10^{-2}$ was used for unit conversion.

### 2.5. Data Analysis and Statistics

The significance of differences in cumulative $N_2O$ emissions among treatments were investigated using one-way ANOVA with LSD test. Differences were considered significant only if $p$ value is less than 0.05. Pearson correlation analysis was used to evaluate the relationships between $N_2O$ fluxes and soil temperature and soil water content in different soil layers. The statistical analyses were conducted using the SPSS 19.0 statistics software (IBM Co., New York, NY, USA) and SigmaPlot 12.5 software packages.

## 3. Results and Discussions

### 3.1. Meteorological Conditions

During the field experimental period from 1 October 2019 to 1 March 2020, air temperature varied in the range of −6.6 to 24.1 °C, presenting a parabolic trend (Figure 2). On 30 November 2019, the daily air temperature was below 0 °C for the first time and subsequently the air temperature decreased in a fluctuating way. The minimum daily air temperature was −6.60 °C on 11 February 2020. After 18 February 2020, daily air temperature was steadily higher than 0 °C. The freezing period was from 30 November 2019 to 18 February 2020, and during the most time of the freezing period, the daytime temperature stayed above 0 °C, and dropped below 0 °C at night. The statistical results showed that the winter of 2019 was the second warmest winter at the experiment site with a negative accumulated temperature of −64.6 °C, compared to the long time mean negative accumulated temperature of −228.32 °C (1960–2019). Total precipitation was 24.3 mm from nine rainfall and two snowfall events, and the highest daily rainfall was about 7.8 mm on 15 February 2020. Due to the warm winter, there was almost no snow cover and the maximum soil frozen depth was 21 cm on 19 February 2020.

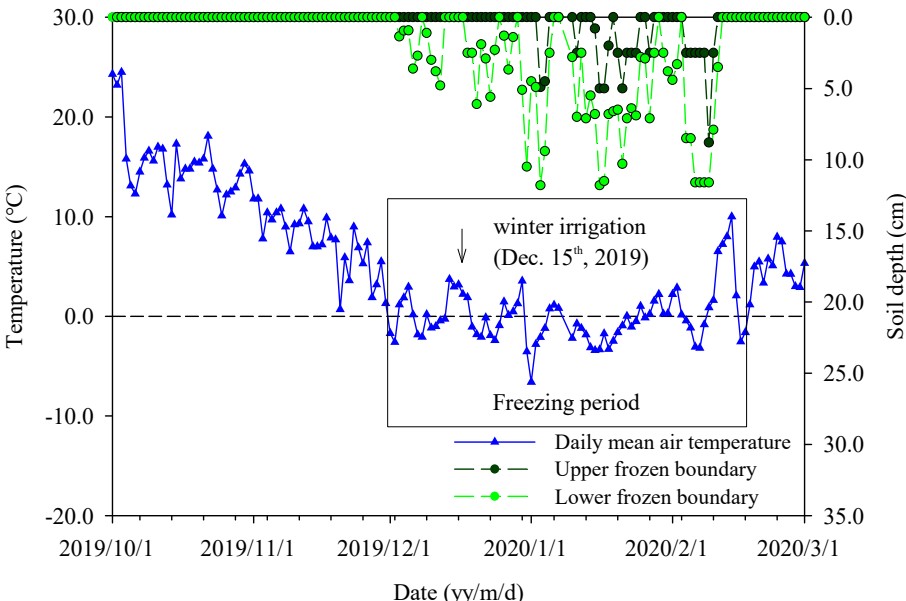

**Figure 2.** Dynamics of air temperature and soil frost depth in the field from 1 October 2019 to 1 March 2020.

### 3.2. Soil Temperature and Soil Water Content

Soil temperature closely followed air temperature and the trends were similar in different soil depth (Figure 3). With the increase of soil depth, soil temperature was increased and the variation of soil temperature was reduced. During the freezing period, soil temperatures at 5, 10, 20, 40 and 60 cm depths varied in the range of −1.5–4.5 °C,

−0.5–4.9 °C, 0.8–4.5 °C, 1.9–6.7 °C and 2.5–9.0 °C, respectively. Diurnal freezing–thawing cycles mainly occurred from middle December 2019 to the end of February 2020 in the 0–10 cm soil. Soil below 20 cm depth remained unfrozen throughout the observation period.

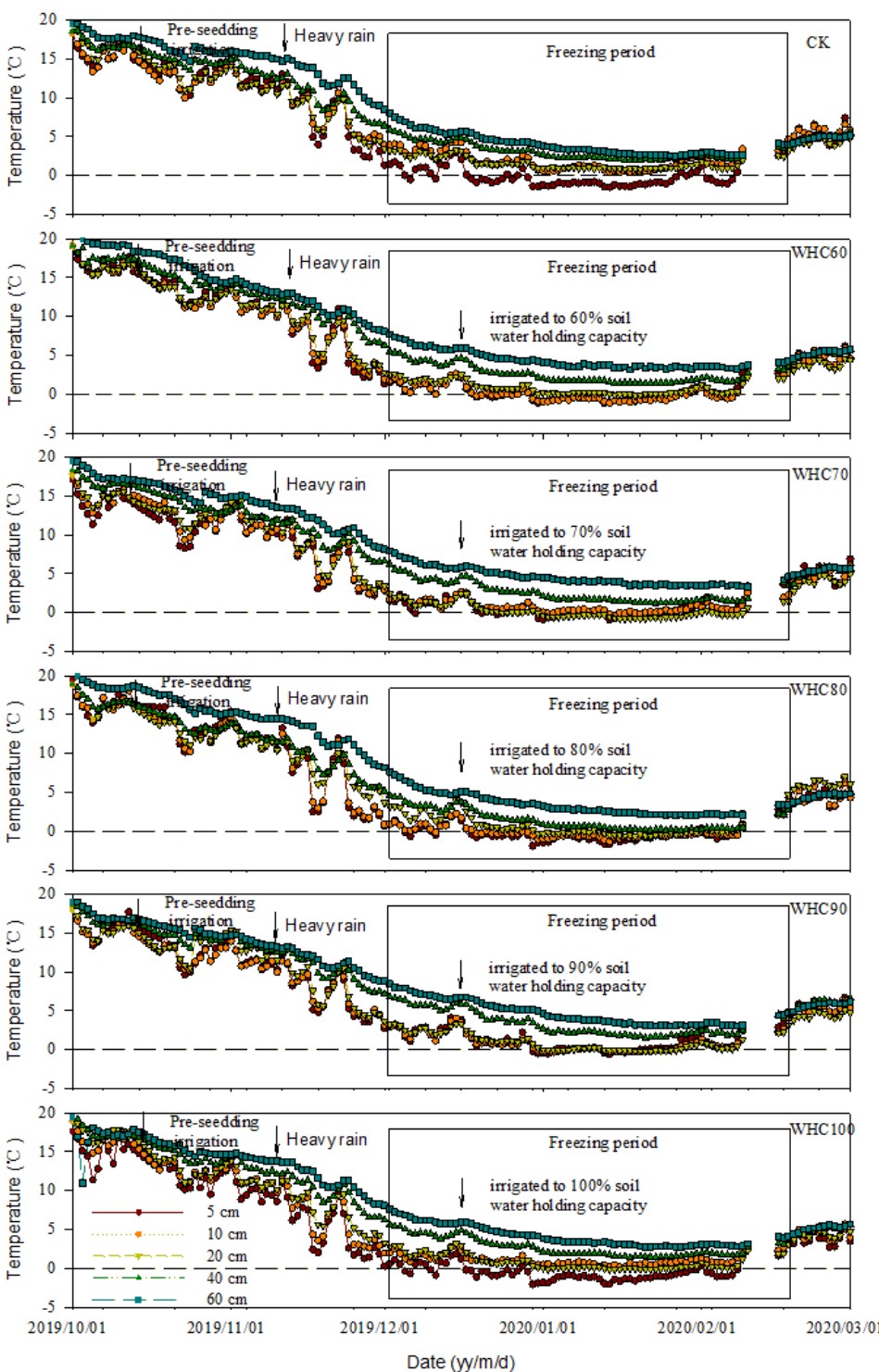

**Figure 3.** Soil temperature at 5, 10, 20, 40 and 60 cm depths from different treatments. Arrows indicate the occurrences of pre-seeding irrigation (17 October 2019), heavy rain (8 November 2019) and winter irrigation (15 December 2019). CK refers to the treatment without winter irrigation. WHC60, WHC70, WHC80, WHC90 and WHC100 refer to the treatments with winter irrigations where soil in 0–30 cm was irrigated to 60%, 70%, 80%, 90% and 100% of the soil water holding capacity on 15 December 2019, respectively.

The dynamics of soil water content are shown in Figure 4. Due to the high evaporation caused by the high wind velocity in the study area, the soil water contents in 0–30 cm soil layer decreased quickly, reaching about 20 $cm^3 \cdot cm^{-3}$ on 15 December 2019. The soil water content in 40–60 cm soil layer was relatively stable. After two days from winter irrigation, the soil water contents of the 0–30 soil layer in WHC60–100 treatments increased to 22.9, 25.30, 30.40, 31.90 and 34.70 $cm^3 \cdot cm^{-3}$, respectively, while the soil water content from CK remained at about 20.50 $cm^3 \cdot cm^{-3}$. Afterwards, the soil water content in all treatments showed a fluctuant reduction trend again. Intermittent rainfalls from 15 February to 1 March improved the soil water content, but the influence mainly focused on the 0–10 cm soil layer and persisted for a short period of time.

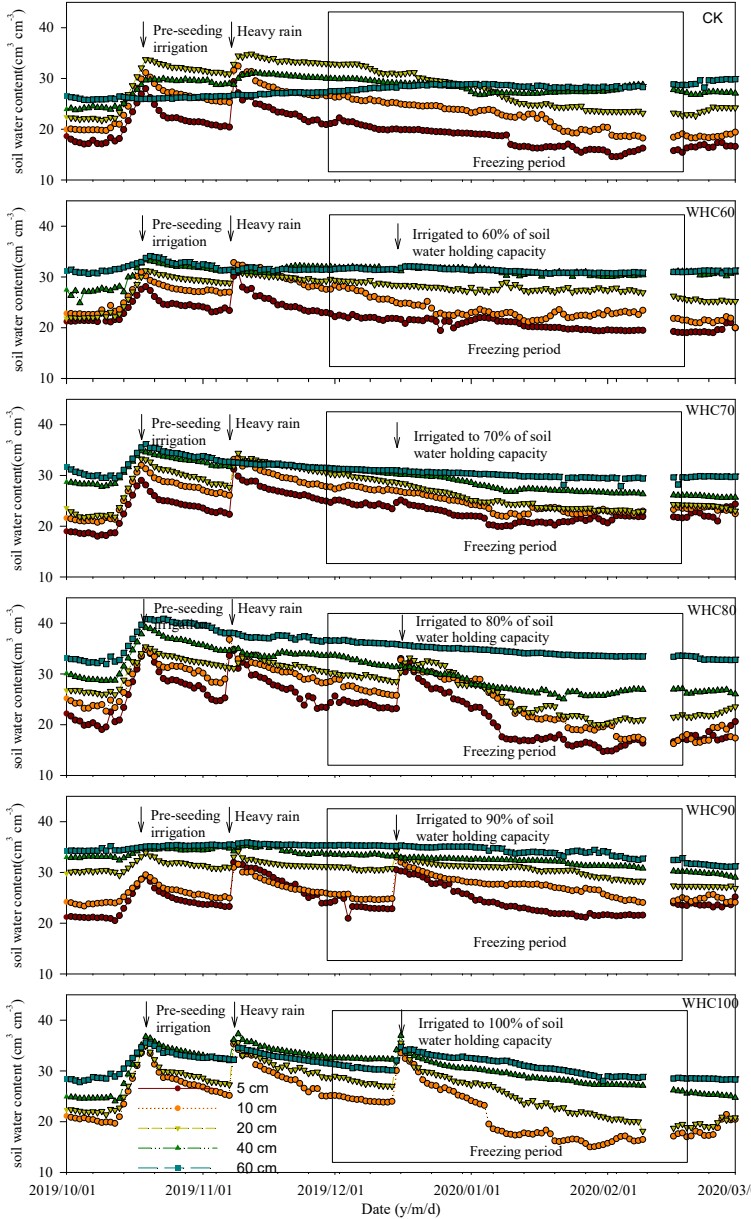

**Figure 4.** Soil water content at 5, 10, 20, 40 and 60 cm depths from different treatments. Arrows indicate the occurrences of pre-seeding irrigation (17 October 2019), heavy rain (8 November 2019) and winter irrigation (15 December 2019). CK refers to the treatment without winter irrigation. WHC60, WHC70, WHC80, WHC90 and WHC100 refer to the treatments with winter irrigations where soil in 0–30 cm was irrigated to 60%, 70%, 80%, 90% and 100% of the soil water holding capacity on 15 December 2019, respectively.

### 3.3. Soil N$_2$O Emission

As shown in Figure 5 and Table 2, high N$_2$O emission was observed on 26 October 2019, reaching about 58 μg·m$^{-2}$·h$^{-1}$. The peak emission lasted only a few days, followed by a sharp drop on 9 November 2019. From 1 December 2019, N$_2$O emissions decreased below 0 μg·m$^{-2}$·h$^{-1}$, when the temperature dropped down to 0 °C. After 18 February 2020, the soil N$_2$O fluxes increased slightly due to the temperature uprising, reaching 12.6 μg·m$^{-2}$·h$^{-1}$ on average. Pearson correlation analysis showed that the correlation between soil N$_2$O flux and soil temperature were moderate with correlation coefficients ranging between 0.62 and 0.68 (Table 3).

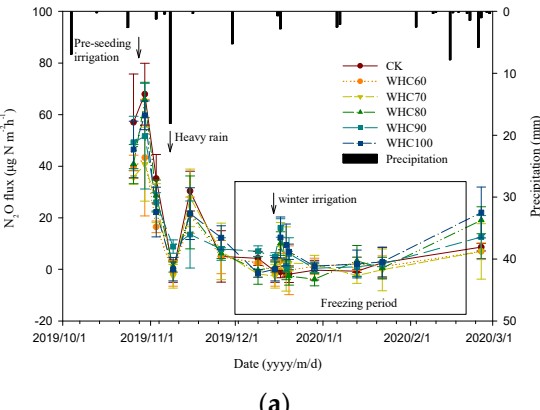

(a)

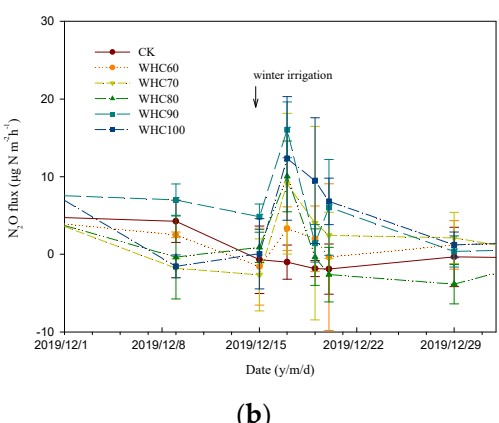

(b)

**Figure 5.** Dynamic changes of soil N$_2$O emission in a winter wheat field (**a**) and a partial enlargement drawing (**b**). Arrows indicate the occurrences of pre-seeding irrigation (17 October 2019), heavy rain (8 November 2019) and winter irrigation (15 December 2019). The bar plot placed in the upper part is precipitation. CK refers to the treatment without winter irrigation. WHC60, WHC70, WHC80, WHC90 and WHC100 refer to the treatments with winter irrigations where soil in 0–30 cm was irrigated to 60%, 70%, 80%, 90% and 100% of the soil water holding capacity on 15 December 2019, respectively.

**Table 2.** Soil N$_2$O emission from different treatments at different sampling time.

| | CK | WHC60 | WHC70 | WHC80 | WHC90 | WHC100 |
|---|---|---|---|---|---|---|
| | | | 2019.10.17 Pre-seeding irrigation | | | |
| 2019.10.26 | 57.0 + 18.7 a | 39.8 + 4.5 a | 35.7 + 2.8 a | 40.8 + 7.6 a | 49.2 + 10.1 a | 46.5 + 10.9 a |
| 2019.10.30 | 67.9 + 12.0 a | 43.3 + 22.6 a | 40.8 + 14.4 a | 65.7 + 6.9 a | 51.6 + 20.5 a | 59.8 + 5.6 a |
| 2019.11.03 | 35.2 + 9.4 b | 16.5 + 2.3 a | 25.7 + 7.6 ab | 28.6 + 6.0 ab | 25.9 + 3.7 ab | 22.2 + 9.6 ab |
| | | | 2019.11.08 Heavy rain | | | |
| 2019.11.09 | −0.7 + 4.3 a | −1.6 + 5.0 a | −2.7 + 4.6 a | 0.9 + 1.9 a | 8.8 + 2.6 b | 0.1 + 4.5 a |
| 2019.11.15 | 30.4 + 7.7 a | 21 + 7.0 a | 27.7 + 11.2 a | 22.1 + 14.1 a | 13.5 + 13.0 a | 21.6 + 10.0 a |
| 2019.11.26 | 5.0 + 10.0 a | 4.8 + 6.4 a | 6.9 + 11.0 a | 6.3 + 2.1 a | 7.9 + 4.4 b | 12.3 + 4.6 a |
| 2019.12.09 | 4.3 + 2.7 b | 2.5 + 0.4 a | −1.8 + 1.2 a | −0.4 + 5.4 a | 7.0 + 2.1 a | −1.6 + 1.5 a |
| | | | 2019.12.15 Winter irrigation | | | |
| 2019.12.15 | −0.7 + 4.3 a | −1.6 + 5.0 a | −2.7 + 4.6 a | 0.9 + 1.9 a | 4.8 + 1.6 b | 0.1 + 4.5 a |
| 2019.12.17 | −1.0 + 2.2 a | 3.3 + 2.8 ab | 9.1 + 9.1 ab | 10.0 + 4.6 ab | 16 + 3.6 a | 12.3 + 8 b |
| 2019.12.19 | −1.8 + 1.0 a | 2.0 + 4.2 a | 4.0 + 12.5 a | −0.4 + 3.6 a | 1.4 + 2.4 a | 9.5 + 8.1 a |
| 2019.12.20 | −1.9 + 3.2 a | −0.4 + 9.5 a | 2.4 + 3.0 a | −2.6 + 3.5 a | 6.0 + 6.2 a | 6.8 + 3.0 a |
| 2019.12.29 | −0.3 + 3.8 a | 1.2 + 3.1 a | 2.1 + 3.3 a | −3.9 + 2.5 a | 0.4 + 2.0 a | 1.2 + 1.7 a |
| 2020.1.13 | −0.7 + 0.5 a | 2.2 + 2.6 | −2.4 + 3.0 a | 3.3 + 6.0 a | 1.0 + 3.7 a | 2.1 + 5.4 a |
| 2020.1.22 | 2.5 + 5.9 a | 1.0 + 4.2 a | −0.2 + 8.1 a | 0.5 + 4.2 a | 2.1 + 1.0 a | 2.9 + 5.8 a |
| 2020.2.16 | 8.6 + 1.5 a | 7.0 + 3.2 a | 7.0 + 10.7 a | 19 + 5.2 a | 12.5 + 8.3 a | 22 + 9.9 a |

Note: Data is expressed as mean ± standard error form (*n* = 3). Values with same letters in the same line are of no significant difference (*p* > 0.05), those with different letters are of significant or extreme difference (*p* < 0.05). CK refers to the treatment without winter irrigation. WHC60, WHC70, WHC80, WHC90 and WHC100 refer to the treatments with winter irrigations where soil in 0–30 cm was irrigated to 60%, 70%, 80%, 90% and 100% of the soil water holding capacity on 15 December 2019, respectively.

**Table 3.** Pearson correlation between $N_2O$ and soil temperature or moisture in different depth in winter wheat field from 1 October 2019 to 1 March 2020.

| | $N_2O$ | T (5 cm) | T (10 cm) | T (20 cm) | T (40 cm) | T (60 cm) | SM (5 cm) | SM (10 cm) | SM (20 cm) | SM (40 cm) | SM (60 cm) |
|---|---|---|---|---|---|---|---|---|---|---|---|
| $N_2O$ | 1 | 0.667 ** | 0.656 ** | 0.662 ** | 0.643 ** | 0.616 ** | 0.246 * | 0.250 * | 0.266 * | 0.165 | 0.081 |

Note: T, SM are abbreviations for soil temperature and soil moisture, respectively. **. Correlation is significant at the 0.01 level (2-tailed).*. Correlation is significant at the 0.05 level (2-tailed).

In soil, $N_2O$ was produced mainly via microbial transformations such as nitrification and denitrification. During winter, low temperatures restricted these microbial activities, causing the low $N_2O$ emissions. Peng et al. [24] and Hu et al. [25] also reported that almost no $N_2O$ was detected during winter in a temperate forest soil and an alpine shrub meadow soil, respectively. In addition, when the soil temperature drops below 0 °C, the decreased availability of soil water restricts soil $N_2O$ formation [26], besides, the formation of ice layer blocks $N_2O$ emissions from soil. Chen et al. [23] also reported that strong correlations ($p < 0.01$) were found between $N_2O$ flux and air or soil temperature during winter. With the rise of ambient temperature, microbial activities in soil increases, accelerating nitrogen cycling and promoting the formation of $N_2O$ in soil.

After two days of winter irrigation, the $N_2O$ fluxes in WHC60 – 100 were increased by 4.3, 10.1, 13.3, 17.0 and 11.0 $\mu g \cdot m^{-2} \cdot h^{-1}$, respectively, compared to the $N_2O$ flux in CK (Table 2). The analysis of variance showed the $N_2O$ fluxes from WHC90 and WHC100 were significantly higher than that from CK treatment (with the $p = 0.011$ and 0.036, respectively) and the difference among other treatments were not significant. Afterwards, soil $N_2O$ fluxes from winter irrigation treatments decreased to 0 $\mu g \cdot m^{-2} \cdot h^{-1}$ again. Pearson correlation analysis showed that the correlation between soil $N_2O$ flux and soil water content was poor with correlation coefficients ranging between 0.08 and 0.25 (Table 3).

Soil moisture was one of the crucial drivers of temporal variability of soil–atmosphere N gas fluxes through the regulation of diffusion conditions, oxygen availability and the soil microbes' respiration chain [13]. As shown in Figure 6, soil emitted more $N_2O$ when the soil water content of top soil layer ranged between 26–31 $cm^3 \cdot cm^{-3}$ (equivalent to 62–73% WFPS), which was in line with the results obtained by Hamamoto et al. [27], who reported that when WFPS is between 50% and 80%, high $N_2O$ emissions were measured. Hung and Whalen [28] also reported that greatest production of $N_2O$ may occur at moderate gas diffusivities (e.g., oxic zones and anoxic microsites coexisted), and excessive moisture condition, usually at >60–70% WFPS, may suppress $N_2O$ emissions due to a shift toward complete denitrification. Klefoth et al. [29] also reported almost all introduced $N_2O$ was reduced to $N_2$ in a soil at 90% WFPS. Chen et al. [23] reported anaerobic conditions can form immediately after soil waterlogging because the $O_2$ diffusion rate can be reduced by 10,000-fold in wet soils, and subsequently $N_2O$ was used as the electron acceptor and further reduced to $N_2$ [30].

It is found that no significant relationship between $N_2O$ emissions and soil moistures. Similar result was obtained by Wang et al. [11] and they explained low temperature or both low temperature and low effective soil water content may be the restriction factors. In our experiment, low temperatures and low soil inorganic nitrogen content may limit the effect of soil water on $N_2O$ emissions, and then weakened their correlations. During the experiment, the soil $NO_3^-$ concentrations varied in the range of 0.25–3.04 mg/kg. Generally, high $N_2O$ fluxes occurred when the $NO_3^-$ concentrations were higher than 3 mg/kg [27]. The soil $NH_4^+$ concentrations were also at a low level, ranging between 0.05 and 2.06 mg/kg. According to the de novo production theory, an inadequate available nitrogen supply has long been considered to be the bottleneck that limits nitrification and denitrification processes in cold regions [31].

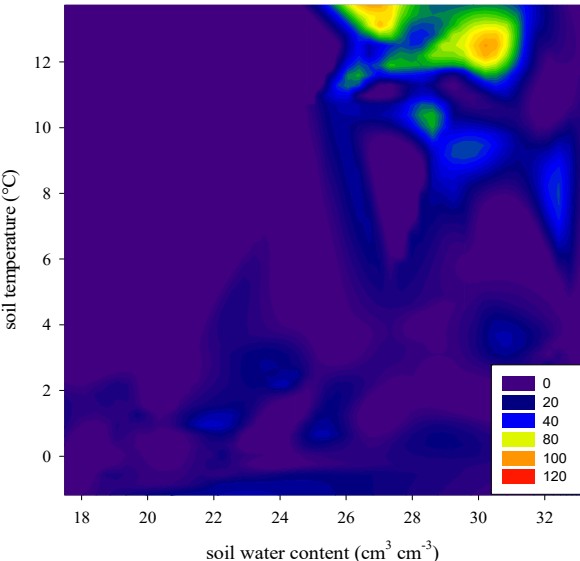

**Figure 6.** Relationship between nitrous oxide ($N_2O$) flux and soil temperature and soil water content in top soil layers.

After winter irrigation (15 December 2019 to 18 February 2020), the average $N_2O$ emission in CK and WHC60–100 were 3.6, 3.4, 4.1, 4.6, 6.8 and 7.4 $\mu g \cdot m^{-2} \cdot h^{-1}$, respectively, and the cumulative emissions during this period in CK and WHC60–100 were 28.0, 36.4, 31.2, 60.8, 71.1 and 110.2 g N $ha^{-1}$, respectively (Figure 7). The cumulative emissions in WHC60–100 were 1.3, 1.1, 2.2, 2.5 and 3.9 times higher than that of CK, respectively. The analysis of variance showed the cumulative emission in WHC100 was significantly higher than that of CK ($p = 0.026$), WHC60 ($p = 0.042$) and WHC70 ($p = 0.031$) and the differences among other treatments were not significant.

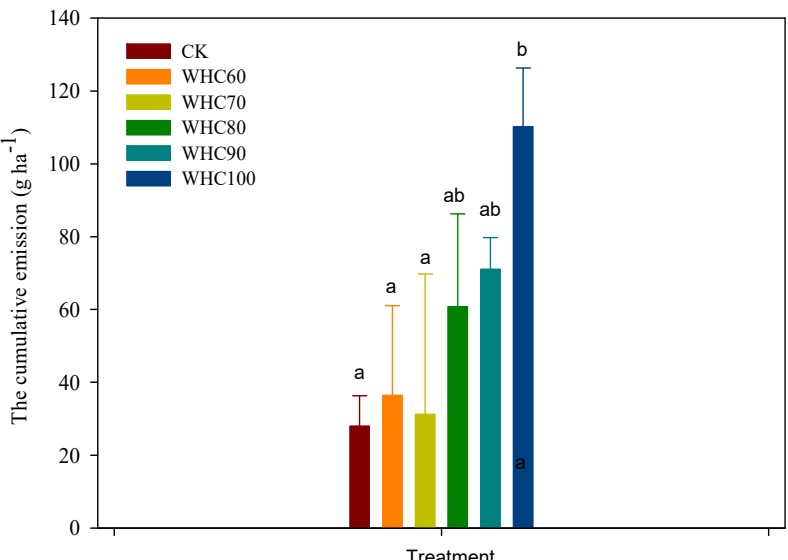

**Figure 7.** The cumulative emissions in CK and WHC60–100 from 15 December 2019 to 18 February 2020. Values with same superscript letters are of no significant difference ($p > 0.05$), those with different letters are of significant or extreme difference ($p < 0.05$). CK refers to the treatment without winter irrigation. WHC60, WHC70, WHC80, WHC90 and WHC100 refer to the treatments with winter irrigations where soil in 0–30 cm was irrigated to 60%, 70%, 80%, 90% and 100% of the soil water holding capacity on 15 December 2019, respectively.

The results showed that winter irrigation had an impact on soil $N_2O$ emission. The cumulative $N_2O$ emissions after winter irrigation (from 15 December 2019 to 18 February

2020) increased with the increasing of the irrigation amount. Peng et al. [24] revealed that watering soil to a relatively high moisture level before freezing could induce significantly higher cumulative emissions of $N_2O$ during multiple freezing–thawing cycles. Juan et al. [32] reported that more $N_2O$ emissions were observed in higher soil moisture and lower freeze temperature treatment during thawing, in which more soil water froze and the expansion of ice crystals could induce larger destruction of soil lattices and be lethal to many microorganisms, releasing more substrates for microbial activities.

However, the enhancement of $N_2O$ emission by winter irrigation in our experiment was relatively constrained, compared to the results obtained by Teepe et al. [21] and Koponen and Martikainen [33] through laboratory experiments. Koponen and Martikainen [33] reported that $N_2O$ production in higher soil water content treatment was 14–106 times higher than that in lower soil water content treatment. The limit $N_2O$ emission enhancement could be attributed to the N-limited condition.

The low inorganic nitrogen content in soil also restricted the $N_2O$ emission in early spring and no obvious emission peaks were observed. Ludwig et al. [34] reported that little or no $N_2O$ was released during thawing and they attributed the phenomenon to the absence of $NO_3^-$ in Russian tundra soils. Regina et al. [35] also reported that the spring thawing in the northern peat field did not induce especially large $N_2O$ emissions compared to the results in the southern peat field, and the soil inorganic nitrogen content was low in the northern peat field. Müller et al. [36] and de Bruijn et al. [37] reported high soil $N_2O$ emissions during spring-thaw and attributed the pulse of $N_2O$ emission to de novo production by denitrification due to rapidly increasing soil temperature, moisture and microbial activity [38] or a physical release of $N_2O$ that was trapped beneath a frozen soil layer [24]. In their experiment, water from snow or ice melt rapidly created an anaerobic environment during the melting period in spring, stimulating denitrification as well as gaseous $N_2O$ emissions. However, since the experiment in this study was conducted in a warm winter, there's no steady frozen soil layer or rapid increase of soil temperature and moisture; therefore, neither physical pulse release of $N_2O$ from beneath the frozen soil layer nor the denitrification conditions contributed to a pulse release.

## 4. Conclusions

The results of our experiment indicated that winter irrigation had an impact on $N_2O$ emissions in the winter wheat field. More $N_2O$ emitted when the soil water content ranged between 26–31 $cm^3 \cdot cm^{-3}$ (equivalent to 62–73% WFPS). The cumulative $N_2O$ emission after winter irrigation in WHC60–100 were 1.1–3.9 times higher than that of CK, indicating that the cumulative $N_2O$ emission were risen by coping the progressively increase of irrigation volumes. Pearson correlation analysis showed that the correlation between soil $N_2O$ flux and soil temperature ranged between 0.62 and 0.67while the correlation between soil $N_2O$ flux and soil water content was ranged between 0.08 and 0.25, indicating the soil $N_2O$ flux has a moderate correlation with soil temperature, but is poorly related to the soil moisture during the warm winter season. To further explore the timing and amount of winter irrigation on soil $N_2O$ emission, the combination of laboratory simulation experiments and long-term field investigation with high monitoring frequency should be carried out in the future.

**Author Contributions:** Conceptualization, Y.A. and Z.G.; methodology, Y.A. and Z.G; formal analysis and investigation: Y.A., Z.G. and K.L.; writing—original draft preparation, Y.A.; writing—review and editing, Z.G., J.X. and Q.W.; supervision, X.J.; funding acquisition, Z.G. All authors have read and agreed to the published version of the manuscript.

**Funding:** This work was supported by the National Natural Science Foundation of China [grant number 51909064], the Project funded by China Postdoctoral Science Foundation [grant number 2019M651684] and the Belt and Road Special Foundation of the State Key Laboratory of Hydrology-Water Resources and Hydraulic Engineering, Hohai University [grant number 2019490411].

**Institutional Review Board Statement:** Not applicable.

**Informed Consent Statement:** Not applicable.

**Data Availability Statement:** Not applicable.

**Conflicts of Interest:** The authors declare no conflict of interest.

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
