# Peer review of "Enhanced N2O Emissions from Winter Wheat Field Induced by Winter Irrigation in the North China Plain"

_agronomy, doi:10.3390/agronomy12040955_

Round 1
Reviewer 1 Report
The manuscript agronomy-1499646, entitled “Enhanced N2O emissions from winter wheat field induced by winter irrigation in the North China Plain” submitted by An et al. report the results of a field experiment were the effect of winter irrigation on N2O emissions were assessed. In particular, the authors tested 5 different irrigation water volumes, corresponding to 5 levels of soil water hold capacity, plus an untreated control during one winter wheat cropping season in North China Plain. During the experiment, ancillary measures like soil moisture and temperature of the soil layer 0-60 were monitored.
Considering the importance of field experiment-based study in order to clarify the effect of winter irrigation to support winter wheat to N2O emission, I believe that the manuscript is of potential interest for readers of “Agronomy” and fall within its scope.
In general, the experimental activity was carried out following a strict scientific logic and according to widely used methods which have made it possible to obtain reliable results. The manuscript style is fine and only small changes are needed to improve language/writing.
Anyway, three main limitations are present in this experiment/manuscript: 1) experiment was carried out only during one cropping season; at least two years of experiments are needed to achieve solid and reliable results. So, for the future, I suggest to the authors to provide a larger dataset based on two years of observation. 2) Only a static chamber per plot not is enough to study a phenomenon that is affected by soil structure that can change a lot inside a plot. 3) Pearson correlation that did you performed showed a significant effect from the temperature but the correlation was very poor, so, temperature effect on N2O emission was not so clear to make correct speculations on this parameter. However, the effort made and the contribution made by this research must be appreciated.
Abstract: It is fine and only small changes are needed. I suggest adding more numeric/percentage numbers.
Keywords: they are fine.
Introduction: only small changes are needed.
Materials and methods: they are ok but I suggest adding information regarding soil water hold capacity determination and cumulative emissions calculation.
Results and Discussion: they are fine but I suggest reversing Table 2 and Figure 4, deleting figure 5 and simplifying table 3. Try to discuss the data immediately after you describe them. Improve the discussion with more cases and by better explaining the mechanisms taking place in the soil.
Conclusion: it is based on the achieved results.
My specific comments, which I hope will help the authors to improve the manuscript, are reported in the attached files.

Reviewer 2 Report
This manuscript is dealing with the effect of winter irrigation on N2O emission on winter wheat. The whole manuscript is well written. The introduction reveals the background of the experiment. The methods are well described and understandable, the results are also well presented. All conclusions are supported by the results.
This experiment has mainly a theoretical importance, so the potential readers are scientists. For practitioners this topic is not significant.
The manuscript is suitable for publication in its present form.
Reviewer 3 Report
Dear authors, please take into account the suggestions I have made to you (in the attached file) in order to improve the scientific value of this manuscript.

Reviewer 4 Report
First, I have to say that the research was performed during only one season. This cannot give reliable conclusions.Second, the experimental design not stated clearly. (1) As the pre-sowing irrigation was applied, what's soil moisture before winter irrigation?(2)Gas samples were collected once every two weeks, I cannot agree that it is a reasonable design.
Third, the freeze-thaw cycling has significant effects on N2O emission, while too few samples were collected during the freeze-thaw cycle.
Round 2
Reviewer 4 Report
- Please explain treatment code in figures in detail.
- I suggested that the correlation between N2O and soil moisture and temperature should be analyzed.
